# Anti-Parkinson Effects of *Holothuria leucospilota*-Derived Palmitic Acid in *Caenorhabditis elegans* Model of Parkinson’s Disease

**DOI:** 10.3390/md21030141

**Published:** 2023-02-23

**Authors:** Tanatcha Sanguanphun, Sukrit Promtang, Nilubon Sornkaew, Nakorn Niamnont, Prasert Sobhon, Krai Meemon

**Affiliations:** 1Department of Anatomy, Faculty of Science, Mahidol University, Rama VI Road, Bangkok 10400, Thailand; 2Molecular Medicine Program, Multidisciplinary Unit, Faculty of Science, Mahidol University, Rama VI Road, Bangkok 10400, Thailand; 3Department of Chemistry, Faculty of Science, King Mongkut’s University of Technology Thonburi, Bang Mod, Bangkok 10140, Thailand; 4Chemistry Program, Department of Science, Faculty of Science and Technology, Bansomdejchaopraya Rajabhat University, Bangkok 10600, Thailand; 5Center for Neuroscience, Faculty of Science, Mahidol University, Rama VI Road, Bangkok 10400, Thailand

**Keywords:** Parkinson’s disease, palmitic acid, sea cucumber, *Holothuria leucospilota*, dopaminergic neuron, α-synuclein

## Abstract

Parkinson’s disease (PD) is the second most common neurodegenerative disease which is still incurable. Sea cucumber-derived compounds have been reported to be promising candidate drugs for treating age-related neurological disorders. The present study evaluated the beneficial effects of the *Holothuria leucospilota* (*H. leucospilota*)-derived compound 3 isolated from ethyl acetate fraction (HLEA-P3) using *Caenorhabditis elegans* PD models. HLEA-P3 (1 to 50 µg/mL) restored the viability of dopaminergic neurons. Surprisingly, 5 and 25 µg/mL HLEA-P3 improved dopamine-dependent behaviors, reduced oxidative stress and prolonged lifespan of PD worms induced by neurotoxin 6-hydroxydopamine (6-OHDA). Additionally, HLEA-P3 (5 to 50 µg/mL) decreased α-synuclein aggregation. Particularly, 5 and 25 µg/mL HLEA-P3 improved locomotion, reduced lipid accumulation and extended lifespan of transgenic *C. elegans* strain NL5901. Gene expression analysis revealed that treatment with 5 and 25 µg/mL HLEA-P3 could upregulate the genes encoding antioxidant enzymes (*gst-4*, *gst-10* and *gcs-1*) and autophagic mediators (*bec-1* and *atg-7*) and downregulate the fatty acid desaturase gene (*fat-5*). These findings explained the molecular mechanism of HLEA-P3-mediated protection against PD-like pathologies. The chemical characterization elucidated that HLEA-P3 is palmitic acid. Taken together, these findings revealed the anti-Parkinson effects of *H. leucospilota*-derived palmitic acid in 6-OHDA induced- and α-synuclein-based models of PD which might be useful in nutritional therapy for treating PD.

## 1. Introduction

One of the fast-rising neurological disorders in the world is Parkinson’s disease (PD) which is estimated to afflict approximately 12 million people by 2040 [1]. The loss of dopaminergic (DAergic) neurons in substantia nigra pars compacta (SNc) and aggregation of insoluble proteins within Lewy bodies (LB) and Lewy neurites have been considered as pathological hallmarks of PD [2,3]. Genetic mutation, environmental toxins and aging contribute to the production of excessive reactive oxygen species (ROS) in SNc neurons which is responsible for their demise [4]. ROS mainly compromises H_2_O_2_, O_2_^−^, and OH^−^ which can be scavenged by antioxidant enzymes [5]. Oxidative stress occurs when there is an imbalance between ROS production and cellular antioxidant activity. The excessive ROS ultimately contributes to oxidative stress-induced neurodegeneration due to damages of macromolecules (DNA, proteins and lipid) and cellular organelles [6]. The aggregation of α-synuclein has been found as the main component within the LB inclusions [7]. Misfolded/toxic α-synuclein aggregation causes intracellular dysregulations of proteostasis, synapsis, mitochondrial function, calcium signaling, redox homeostasis, and oxidative stress [8]. As a result of degeneration of motor-controlling SNc DAergic neurons in the striatum from these pathological causes, PD patients suffer not only motor disabilities but also non-motor symptoms, including neuropsychiatric symptoms and dementia [9]. Unfortunately, there is still no efficacious treatment to mitigate progressive PD, whilst the current treatments for this disease offer only symptomatic relief. Therefore, discoveries of therapeutics that can inhibit oxidative-induced DAergic neurodegeneration and α-synuclein aggregation should be the targets for curing or at least slowing down PD progression.

In recent years, natural products from animals and plants have been developed as new therapeutic drugs for PD [10]. Sea cucumbers are invertebrate marine animals which are commonly used as traditional medicine in Asian countries [11], as they contain various bioactive components, including peptides, fucoidan, triterpene glycosides, cerebrosides, fucosylated chondroitin sulfate, phenols, saponin, phospholipids and essential fatty acids [11,12]. Based on their impressive chemical profiles, a wide range of ascribed biological functions of sea cucumbers have been reported, such as anticancer, anti-inflammation, anti-oxidation, anti-diabetes, anti-obesity, improvement of memory and learning, and anti-aging [11,13]. Previous study reported the anti-neurodegenerative action of the crude ethyl acetate fraction of *Holothuria scabra* (*H. scabra*) in *C. elegans* PD model [14]. In addition, *H. scabra*-derived diterpene glycosides have been reported to protect against α-synuclein-induced loss of DAergic neurons by upregulating autophagy mediators (*bec-1, lgg-1* and *atg-7*) [15]. Apart from *H. scabra*, one of the commonly widespread and more abundant species in the Indo-Pacific Ocean is *Holothuria leucospilota* (*H. leucospilota*) which is usually known as the black sea cucumber. Interestingly, there are some studies reporting the beneficial effects of the crude and purified compound from this species. Saponin-rich butanol extract of *H. leucospilota* upregulated stress resistance, decreased aging biomarkers and extended lifespan of *C. elegans* via DAF-16 signaling [16]. Ethyl acetate fraction of *H. leucospilota* (HLEA) and the HLEA-P1 compound or decanoic acids purified from this fraction have been shown to possess neuroprotective effects against DAergic neurodegeneration and α-synuclein toxicity in *C. elegans* PD models [17,18]. Apart from HLEA-P1/decanoic acid, HLEA-P3 compound has been found in the HLEA fraction. The present study hypothesized that HLEA-P3 might confer neuroprotective effects against PD. In this study, *C. elegans* was utilized as a model for studying the PD pathology because it has several advantages, including a short life cycle, eight DAergic neurons with conserved DAergic pathway and genes with human orthologs. Moreover, it is easy to maintain, PD-like phenotypes can be easily generated, and behaviors analysis can be analyzed in laboratory [19,20]. DAergic neurodegeneration was induced by exposing worms with a neurotoxin 6-hydroxydopamine (6-OHDA). 6-OHDA has been widely reported to produce ROS as it enters the DAergic neuron through dopamine transporter to disrupt the neurons by promoting oxidative stress, decreasing detoxify system and activating apoptosis [21]. Additionally, transgenic *C. elegans* strain NL5901, which has human α-synuclein tagged-yellow fluorescent protein (YFP) in the body muscle cells, has been used for a model of α-synuclein aggregation. Consequently, the present study evaluated the effect of HLEA-P3 in these *C. elegans* and found its positive anti-Parkinson potential and the possible molecular mechanism.

## 2. Results

### 2.1. HLEA-P3 Attenuated DAergic Neurodegeneration Induced by 6-OHDA

The loss of DAergic neurons in SNc is the neuropathological feature of PD. In the present study, *C. elegans* were exposed to 6-OHDA that selectively destroyed their DAergic neurons. According to a previous study, four DAergic neurons at cephalic sensilla (CEP) are the primary targets of 6-OHDA [22]. Therefore, the GFP intensity of CEP neurons was measured for scoring the viability of DAergic neurons in *C. elegans*. As shown in Figure 1A, normal transgenic BY250 worms possessed complete four CEP neurons with strong GFP intensity, whereas 6-OHDA-treated worms exhibited incomplete GFP-tagged CEP neurons. The results showed that the GFP-tagged CEP intensity of 6-OHDA-treated and 6-OHDA/DMSO-treated worms significantly decreased to 64.02% and 63.77%, respectively, when compared to normal BY250 worms. These suggested the degeneration of DAergic neurons upon 6-OHDA treatment. Interestingly, the fluorescence intensity of GFP-tagged CEP neurons was significantly increased to 82.50%, 90.56% and 91.54% in 1, 5 and 25 μg/mL after HLEA-P3 treatment, respectively, (*p* < 0.05) when compared to the untreated group. Although 50 μg/mL HLEA-P3 increased the fluorescence intensity of GFP-tagged CEP neurons to 82.08%, compared to the untreated group, the restorative effect was lower than those of 5 and 25 μg/mL HLEA-P3 (Figure 1B). Therefore, 5 and 25 μg/mL HLEA-P3 were chosen for further analysis.

### 2.2. HLEA-P3 Significantly Improved Dopamine-Dependent Behaviors in 6-OHDA-Treated C. elegans

In *C. elegans*, DAergic signaling regulates several behaviors, particularly feeding and chemo-perception [23]. In order to investigate the function of DAergic neurons, well-known dopamine-dependent behaviors including basal slowing and ethanol avoidance behaviors were observed. Basal slowing is a food sensing behavior that occurs when worms decrease their locomotion in the presence of food [23]. When compared to normal worm, the basal slowing rates of 6-OHDA treated and 6-OHDA/DMSO-treated worms were significantly decreased to approximately 67.56% and 69.64%, respectively (*p* < 0.05). Interestingly, treatments with 5 and 25 μg/mL of HLEA-P3 recovered basal slowing rate to 98.03% and 96.87%, respectively, compared with the untreated group (*p* < 0.05) (Figure 2A). Likewise, defective ethanol avoidance behavior was observed in worms treated with 6-OHDA and was recovered by the HLEA-P3 treatment. As shown in Figure 2B, the ethanol avoidance index was about −0.25 and −0.23 in 6-OHDA-treated and 6-OHDA/DMSO-treated worms, respectively, but significantly increased to 0.3 and 0.24 after treatment with 5 and 25 μg/mL of HLEA-P3, respectively (Figure 2B). These results suggested that HLEA-P3 significantly improved dopamine-dependent behaviors which were suppressed by 6-OHDA in *C. elegans* PD model.

### 2.3. HLEA-P3 Reduced Oxidative Stress Induced by 6-OHDA in C. elegans

Given that 6-OHDA exerts neurotoxicity by targeting mitochondria and generating oxidative stress [24]; thus, the effect of HLEA-P3 on the reduction in 6-OHDA-mediated oxidative stress was investigated using 2′,7′-Dichlorodihydrofluorescein diacetate (H_2_DCF-DA) assay. The intracellular ROS levels remarkably increased to 239.58% and 250.60% in 6-OHDA-treated and 6-OHDA/DMSO-treated groups, respectively (*p* < 0.05), compared to normal worms. As expected, treatments with 5 and 25 μg/mL HLEA-P3 significantly reduced intracellular ROS levels to 77.18% and 82.80%, respectively (*p* < 0.05) when compared to the untreated group (Figure 3).

### 2.4. HLEA-P3 Reduced α-Synuclein Aggregation and Improved Thrashing Behavior in Transgenic C. elegans Expressing α-Synuclein

This study investigated the effect of HLEA-P3 on α-synuclein accumulation utilizing *C. elegans* NL5901 strain which expresses YFP-tagged human α-synuclein under a muscle-specific promoter. The results showed that worms treated with 5, 25 and 50 μg/mL of HLEA-P3 exhibited a significant decrease in the YFP intensity to 75.72%, 83.11% and 91.70%, respectively (*p* < 0.05), while 1 μg/mL of HLEA-P3-treated worms showed no significant difference (97.47%, *p* > 0.05). These indicated that levels of α-synuclein aggregation were markedly reduced by 24.28%, 16.89% and 8.3%, respectively (Figure 4A,B). With their pronounced effect, 5 and 25 μg/mL of HLEA-P3 were selected for further experiments. In *C. elegans*, protein aggregation is associated with body bending deficits [25]. Previous studies demonstrated that worms expressing α-synuclein in the body wall muscle exhibited impairment of thrashing behavior which refers to the motility rate of worms in a liquid media [26]. Therefore, this present study investigated whether HLEA-P3 could restore locomotory deficit caused by α-synuclein aggregation. The results showed that normal wild-type worms on day 3 and day 5 had trashing rates at 1.34 and 1.21, respectively. Consistent with a previous report, NL5901 worms showed thrashing deficits with thrashing rates decreased to 0.85 and 0.46 on day 3 and day 5 of adulthood, respectively (*p* < 0.05). Interestingly, impaired thrashing behavior was significantly increased (*p* < 0.05) by treatments with 5 and 25 μg/mL HLEA-P3 to 1.03 and 1.06 for day 3 adult NL5901 worms, respectively, and to 1.02 and 0.87 in day 5 adult worms treated with 5 and 25 μg/mL of HLEA-P3, respectively (*p* < 0.05) (Figure 4C).

### 2.5. HLEA-P3 Reduced Lipid Accumulation in Transgenic C. elegans NL5901 Strain

Alteration of lipid composition has been reported in PD patients and various animal models of PD [23]. In this study, the effect of HLEA-P3 on lipid deposition was measured using Nile Red staining. The results showed that NL5901 worms exhibited lower lipid content than wild-type N2 animals. Nile Red intensity was significantly reduced to 74.97% and 74.19% in NL5901 and DMSO-treated NL5901 worms, respectively (*p* < 0.05). HLEA-P3 treatments at doses of 5 and 25 μg/mL significantly reduced lipid deposition to 54.13% and 54.87%, respectively (*p* < 0.05) (Figure 5A,B). Silencing of stearoyl-CoA desaturase-1 (SCD-1), a key enzyme in fatty acid metabolism, was shown to reduce lipid content and α-synuclein-induced neurotoxicity [27]. This study examined whether HLEA-P3 downregulated *fat-5*, *fat-6* and *fat-7* (SCD-1 homolog) in NL5901 worms (Figure 5C). The qRT-PCR results showed that 25 μg/mL HLEA-P3 significantly downregulated *fat-5* mRNA expression (*p* < 0.05). A total of 5 and 25 μg/mL of HLEA-P3 slightly downregulated *fat-7* but with no significant differences. However, HLEA-P3 did not alter mRNA expression of *fat-6*.

### 2.6. HLEA-P3 Significantly Prolonged Lifespan of C. elegans PD Models

The present study determined the lifespan of 6-OHDA-induced worms compared to normal worms. Similar to the previous report, PD worms exhibited a shorter lifespan when compared to normal N2 wild-type worms. As shown in Figure 6 and Table 1, lifespan of 6-OHDA-treated worms and 6-OHDA/DMSO-treated worms decreased by 31.12% and 30.41%. Treatments with 5 and 25 μg/mL of HLEA-P3 significantly extended lifespan of 6-OHDA-treated worms by 18.98% and 11.71%, respectively, compared to the untreated group (*p* < 0.05). Similarly, mean lifespan of α-synuclein expressing NL5901 worms can be extended by 25.97% and 24.57% after treatments with 5 and 25 μg/mL of HLEA-P3, respectively (*p* < 0.05). The results indicated that HLEA-P3 significantly prolonged lifespan of both 6-OHDA-induced- and α-synuclein-based- *C. elegans* PD models.

### 2.7. HLEA-P3 Upregulated Antioxidant Genes and Autophagic Mediators in C. elegans PD Models

To elucidate whether HLEA-P3 decreases oxidative stress by activating an antioxidant defense mechanism, mRNA expression levels of genes related in detoxification such as *gst-4*, *gst-10* and *gcs-1* were measured using the qRT-PCR method. This study found that expressions of mRNAs of *gst-4*, *gst-10* and *gcs-1* genes were significantly upregulated after treatment with HLEA-P3 compound. The 5 μg/mL HLEA-P3 treatment significantly increased expressions of mRNAs of *gst-10* (1.36-fold) and *gcs-1* (1.70-fold), compared to those of untreated group. The 25 μg/mL HLEA-P3 treatment significantly upregulated mRNAs of *gst-4* (4.67-fold), *gst-10* (1.44-fold). The present study also measured mRNA expression levels of *cat-2*, the homolog of mammalian tyrosine hydroxylase (TH), and found a 1.72-fold increase in mRNA expression in 25 μg/mL HLEA-P3-treated group (Figure 7A).

Autophagic activation is responsible for degradation of large protein debris such as α-synuclein oligomers and fibrils [28]. Therefore, the present study evaluated whether HLEA-P3 reduced α-synuclein aggregation by activating autophagy. mRNA expression levels of autophagic mediators, including *bec-1* (autophagy initiation), *atg-7* (autophagosome formation) and *lgg-1* (autophagosome elongation), were measured, and the present study found that treatment with 5 μg/mL HLEA-P3 significantly upregulated *bec-1* mRNA expression up to 1.86 folds when compared to untreated group. Treatment with a higher concentration of 25 μg/mL HLEA-P3 significantly upregulated mRNA expression levels of *bec-1* (1.70-fold) and *atg-7* (2.43-fold), but not *lgg-1* (Figure 7B).

### 2.8. Chemical Structural Analysis Identified HLEA-P3 as Palmitic Acid

From the structural elucidation by using ^1^H and ^13^C NMR analysis, HLEA-P3 was identified as palmitic acid or hexadecanoic acid [18]. HLEA-P3 is a white powder. The ^1^H-NMR/^13^C NMR spectra showed HR-TOFMS (ES^+^): m/z 279.0796 [M + Na]^+^, calcd for C_16_H_32_O_2_ + Na. ^1^H-NMR (CDCl_3_, 400 MHz): 0.86 (3H, t = 7.2, H-16), 1.23 (24H, m, H-4-15), 1.61 (2H, m, H-3), 2.32 (2H, t, H-2). ^13^C-NMR (CDCl_3_, 100 MHz): 14.1 (C-16), 22.7 (C-15), 24.7 (C-3), 29.0 (C-4), 29.3 C-5, 13), 29.6 (C-6-12), 31.9 (C-14), 34.0 (C-2), 178.4 (COOH). As shown in Appendix A, Peak A at 0.86 ppm indicates the presence of terminal methyl group (CH_3_) attached to the C_15_. Peak B at 1.23 ppm corresponds to a long chain of methylene protons (CH_2_) of the C_4_-C_15_ atoms. Peak C at 1.61 ppm is related to 2 protons attached to the C_3_ atom. Peak D at 2.32 ppm corresponds to the methylene protons (CH_2_) of C_2_ (Appendix A).

## 3. Discussion

In this research, 6-OHDA was used to specifically induce DAergic neuronal damages in worms to mimic neuropathological feature of PD. 6-OHDA-treated worms exhibited PD-related phenotypes, including loss of DAergic neurons, impairment of DA-related behaviors, increased oxidative stress and shortened lifespan. This study demonstrated that *H. leucospilota*-derived palmitic acid has a protective effect against 6-OHDA-induced DAergic neurodegeneration. Pathogenic mechanism implicated in PD involves oxidative stress [5]. 6-OHDA causes DAergic neuronal damages by producing reactive oxygen species [29]. In this study, the H_2_DCF-DA results suggested that *H. leucospilota*-derived palmitic acid relieved oxidative stress induced by 6-OHDA. Clinical studies and experimental PD models reported the deficient antioxidative defense [30]. Interestingly, *H. leucospilota*-derived palmitic acid potentially activated the antioxidative response in 6-OHDA-treated worms. This study revealed that *H. leucospilota*-derived palmitic acid upregulated expressions of detoxification genes including *gst-4*, *gst-10* and *gcs-1.* All these genes play a significant role in phase II detoxification enzymes to protect against oxidative stress and promote lifespan [31]. Treatment with *H. leucospilota*-derived palmitic acid upregulated antioxidant response which eventually suppressed 6-OHDA-induced DAergic cell death, as shown by the restored fluorescence signal of GFP-tagged DAergic neurons and increased mRNA expression of *cat-2*, tyrosine hydroxylase (TH) for dopamine synthesis, implicating DAergic neurons were rescued, while *cat-2* mutants exhibited defective DAergic-dependent behaviors [23]. In the present study, *H. leucospilota*-derived palmitic acid improved basal slowing and ethanol avoidance behaviors in 6-OHDA-treated worms. Recently, palmitic acid-enriched diet induces TH protein and mRNA expression in a mice model [32]. Several studies reported a link between elevated oxidative stress and reduced lifespan [33,34]. The present study reported that 6-OHDA shortened lifespan of *C. elegans*. Interestingly, *H. leucospilota*-derived palmitic acid restored lifespan of 6-OHDA-treated worms. Palmitic acid is a long chain saturated fatty acid that has been shown to be involved with the regulation of longevity pathway [35]. Palmitic acid isolated from *H. scabra* could increase GST-4 expression and improve lifespan in *C. elegans* model [36]. Diets rich in high saturated fatty acid promote lifespan extension in a calorie-restricted mice model [37].

This study also demonstrated the antioxidative property of *H. leucospilota*-derived palmitic acid against 6-OHDA in *C. elegans* PD model. In agreement with our findings, the previous study has reported antioxidative activity of palmitic acid isolated from *Vitex negundo* leaves [38]. Moreover, palmitic acid isolated from plant *Syzygium littorale* exhibits free radical scavenging activities [39]. On the contrary, there are several studies demonstrating cytotoxic effects of palmitic acid in peripheral tissues such as liver and muscles by activating mitochondrial dysfunction, endoplasmic reticulum stress and oxidative stress [40,41,42]. Previous study reported that palmitic acid induces oxidative stress and apoptosis of neurons and astrocytes [43]. The controversial effects of palmitic acid may be due to several factors, such as concentration, duration of treatment, and types of models used in the studies.

Additionally, the present study investigated the protective effect of *H. leucospilota*-derived palmitic acid against α-synuclein toxicity using transgenic *C. elegans* NL5901 overexpressing α-synuclein. Reduced YFP-tagged α-synuclein fluorescence intensity indicates that *H. leucospilota*-derived palmitic acid could decrease α-synuclein aggregation in *C. elegans* PD model. NL5901 worms exhibited pathological features of PD, including motor deficit and impaired longevity. Consistent with the decrease in α-synuclein aggregation, locomotion and lifespan of NL5901 worms were improved by *H. leucospilota*-derived palmitic acid treatment. Large protein debris such as α-synuclein oligomers and fibrils is normally degraded by the autophagy-lysosomal pathway [44]. Impairment of autophagy promoted α-synuclein aggregation which is tightly associated with the pathophysiology of PD [28]. Knockout of *Atg7*, which is an enzyme for autophagosome formation, promotes accumulation of presynaptic α-synuclein in vivo [45]. The present study reported the upregulation of autophagic mediators (*bec-1* and *atg-7)* but not *lgg-1*, by *H. leucospilota*-derived palmitic acid. These suggested that *H. leucospilota*-derived palmitic acid enhanced autophagy at the step of initiation and autophagosome formation rather than autophagosome elongation. Accumulating evidence indicated that autophagy is a crucial process for maintaining cellular homeostasis in response to excessive accumulation of fatty acids, while the blocking of autophagy exacerbated cellular damages and apoptosis [46,47]. A previous study reported that palmitic acid increased autophagic flux by activating protein kinase C signaling pathway [47]. Additionally, palmitic acid activated the formation of autophagic vesicles by upregulating beclin-1 in podocytes [46].

Several cellular and animal models of PD reported that α-synuclein-lipid interaction increases the propensity of α-synuclein aggregation [48,49]. The ratio of lipid/protein and the composition of lipid is important factor for the aggregation of α-synuclein and subsequent cellular stresses [48]. With its lipid-binding motif, α-synuclein has high affinity to bind with membrane lipids, especially unsaturated fatty acid, leading to formation and stabilization of α-synuclein aggregates [50]. Compared to saturated fatty acids, unsaturated fatty acids are more susceptible to lipid peroxidation because they contain double bonds which are easily attacked by ROS [51]. Previously, there are studies reporting that the byproducts of lipid peroxidation potentially enhanced α-synuclein aggregation [52,53]. The composition of fatty acids is regulated by a series of fatty acid elongation and desaturation [54]. Monounsaturated fatty acid undergoes sequential desaturation, resulting in monounsaturated and polyunsaturated fatty acids [54]. SCD-1 is a desaturating enzyme responsible for the conversion of palmitic acid (16:0) and stearic acid (18:0), to unsaturated fatty acids, palmitoleic acid (C16:1, Δ9) and oleic acid (C18:1, Δ9), respectively [55]. Transcriptomic analysis revealed that palmitic acid impacted several signaling pathways including lipid metabolism in neurons [56]. This study demonstrated that *H. leucospilota*-derived palmitic acid downregulated *fat-5* which is an SCD-1 homolog in NL5901 worms expressing α-synuclein. Consistent with our finding, inhibition of SCD-1 reduced α-synuclein toxicity in yeast and human neurons [57]. Genetic silencing of SCD-1 homolog decreased lipid droplets, reduced unsaturated oleic acid level and ameliorated α-synuclein-induced DAergic neurodegeneration in *C. elegans* models [58]. Although the type of lipids in *C. elegans* NL5901 was not investigated in this study, it is possible that *H. leucospilota*-derived palmitic acid might reduce the augmentation of toxic unsaturated fatty acids which then reduced the aggregation propensity of α-synuclein in NL5901 worms. Previous study demonstrated that the body wall extract of *H. leucospilota* reduced fat accumulation by downregulating lipogenesis in the *C. elegans* model of obesity [59]. However, the mechanistic aspects of this event need to be studied in more detail.

Taken together, all results suggested that *H. leucospilota*-derived palmitic acid alleviated neurotoxicity caused by 6-OHDA and α-synuclein aggregation in *C. elegans* PD models. By contrast, it was reported that overconsumption of saturated fatty acids, including palmitic acid, could cause neurodegenerative diseases including PD [60]. However, at low doses, palmitic acid may have beneficial effect causing mild stress that can activate stress response pathway to counteract deleterious damages such as oxidative stress. In this study, the effect of *H. leucospilota*-derived palmitic acid was not in a dose-dependent manner since the high dose (50 μg/mL) was less effective in restoring DAergic neurons and reducing α-synuclein aggregation than the medium doses, while the low dose (1 μg/mL) also showed lower effect. This pattern of effective treatment in optimal doses is under hormesis condition, where high dose has negative effect while low dose shows less or no effect. With particularly optimal doses and conditions used in this study, *H. leucospilota*-derived palmitic acid might act as a mild stressor protecting *C. elegans* from 6-OHDA toxicity and α-synuclein aggregation. In addition, although the purity of HLEA-P3 is at 99.6%, the anti-PD effects might also be synergistic between *H. leucospilota*-derived palmitic acid and other minor/undetectable compounds found in HLEA-P3 (Appendix A). Therefore, further experiments to evaluate and compare between *H. leucospilota*-derived palmitic acid and pure palmitic acid on mechanism of action are aimed to be explored in the future before promoting the use of palmitic acid in PD nutritional therapy.

## 4. Materials and Methods

### 4.1. Strains, Growth Condition and Synchronization of C. elegans

The *C. elegans* strains used in this study were obtained from the *Caenorhabditis* Genetics Center (CGC): N2 (Wild-type), NL5901 [*pkIs2386, unc-54p::α-synuclein::YFP + unc-119(+)*] which expresses YFP-tagged human α-synuclein in body wall muscle. BY250 [*vtIs7; dat-1p::GFP*] which expresses GFP in DAergic neuronal cell bodies and processes was kindly provided by Prof. Dr. Randy Blakely, Florida Atlantic University, United States. All strains were cultured on solid nematode growth medium (NGM) seeded with *Escherichia coli* (*E. coli*) OP50 and maintained at 20 °C. Age-synchronized populations were prepared by exposing gravid adult worms with hypochlorite solution (12% (*v*/*v*) sodium hypochlorite and 10% (*v*/*v*) 1M sodium hydroxide) for 10–12 min. Then, egg pellets were separated by centrifugation at 4000 rpm for 90s, washed three times by M9 buffer and transferred to unseeded NGM plates and incubated overnight at 20 °C. Newly hatched L1 larvae were transferred to an OP50-seeded NGM plate, and allowed to grow to the L3 stage for further use in various assays. All experiments performed in the *C. elegans* were ethically approved by the Faculty of Science, Mahidol University–Institutional Animal Care and Use Committee (MUSC–IACUC) according to the protocol number MUSC60-048-398.

### 4.2. Extraction, Isolation and Chemical Characterization of HLEA-P3 from H. leucospilota Ethyl Acetate Fraction

The procedures for handling the sea cucumbers were ethically performed under the guidelines of MU-IACUC according to the protocol number MUSC60-049-399. The extraction and isolation of *H. leucospilota* compounds were performed, as described previously [18]. Briefly, the black sea cucumber *H. leucospilota* samples were provided by Coastal Fisheries Research and Development Center, Prachuap Khiri Khan, Thailand. The body wall samples were collected and lyophilized using a Supermodulyo-230 freeze dryer. A total of 1.2 kg of the freeze-dried samples were pestled to small powder and macerated with hexane, obtaining the hexane fraction (3.2 g) and residue. Then, the acquired residue was extracted by ethyl acetate to obtain HLEA fraction (3.5 g).

The HLEA fraction was purified using sequential liquid–liquid extraction, silica-gel column chromatography (CC) and Thin Layer Chromatography (TLC). In this study, CC was performed using Merck silica gel 60 (finer than 0.063 mm) and Pharmacia Sephadex LH-20. TLC was carried out using Merck precoated silica gel 60 F_254_ plates. The spots on TLC were detected under the UV light and sprayed with anisaldehyde-sulphuric acid reagent followed by heating. The HLEA fraction was fractionated by CC with MeOH 100% as a solvent on Sephadex LH-20 column. Then, the eluates were examined by the TLC to finally obtain 3 subfractions including fraction EA1 (101 mg), fraction EA2 (135 mg), and fraction EA3 (58 mg). Fraction EA1 was subjected to CC using n-hexane-EtOAc (80:20) to afford compound 1 (HLEA-P1, 25.8 mg). Fraction EA2 (135 mg) was subjected to CC using n-hexane-EtOAc (80:20) to afford compound 2 (HLEA-P2, 20.2 mg) and compound 3 (HLEA-P3, 18.3 mg). Fraction EA3 (258 mg) was isolated by CC using n-hexane-EtOAc (70:30) to yield compound 4 (HLEA-P4, 20.5 mg), compound 5 (HLEA-P5, 15.1 mg) and compound 6 (HLEA-P6, 20.1 mg).

In this study, HLEA-P3 isolated from fraction EA2 with 99.6% purity was studied. The structure of HLEA-P3 was chemically analyzed by ^13^C/^1^H- NMR. ^1^H and ^13^C NMR spectra were recorded using a Bruker AVANCE 400 FT-NMR spectrometer operating at 400 (^1^H) and 100 (^13^C) MHz. The high-resolution mass spectra were obtained using Bruker micrOTOF-QII mass spectrometer.

### 4.3. 6-OHDA-Induced DAergic Neurodegeneration Assay and HLEA-P3 Treatment

Selective loss of DAergic neurodegeneration in *C. elegans* was induced by 6-OHDA exposure. The protocol was performed, as described in the previous study with minor modifications [22]. Synchronized L3 were incubated in 500 μL of inducing solution containing 50 mM 6-OHDA (Sigma, St. Louis, MO, USA), 10 mM ascorbic acid (Sigma, St. Louis, MO, USA), and diluted OP50 for 1 h at 22 °C. During the induction, the solution was gently mixed every 10 min. After that, the worms were washed by M9 buffer at least three times or until the supernatant was clear. Then, 6-OHDA-treated worms were transferred to OP50-seeded NGM plates containing 1, 5, 25 and 50 μg/mL of HLEA-P3 and 50 μM 5-fluoro-2′-deoxyuridine (FUdR). In the untreated control group, worms were fed with OP50 mixed with 1%DMSO (*v*/*v*).

### 4.4. Quantitative Analysis of the Viability of DAergic Neurons

After 6-OHDA exposure and 72 h of HLEA-P3 treatment, worms were collected for observing DAergic neurons under fluorescence microscopy. Worms were washed, put onto a 2% agar slide. Then, a drop of 30 mM sodium azide was added to immobilize worms followed by covering with a coverslip. Fluorescence images were taken using a fluorescence microscope (BX53; Olympus Corp., Tokyo, Japan). The viability of dopaminergic neurons was quantified by measuring the GFP-tagged DAergic neurons. Fluorescence intensity of DAergic neurons of each worm was measured using ImageJ software (National Institute of Health, NIH, Bethesda, MD, USA).

### 4.5. Assay for Basal Slowing Response Behavior

After 6-OHDA exposure and 72 h of HLEA-P3 treatment, worms were washed by M9 buffer three times to remove any residual bacteria attached to their bodies. Then, worms were transferred to the assay plates which are NGM plates (no food) and OP50-seeded NGM plates (with food). Worms were allowed to recover for 5 min, and their body bending were recorded for 20s. The basal slowing response is calculated following the formula reported in previous study [17]. The basal slowing rate = 100 − locomotory rate (%), where the locomotory rate (%) = [rate of bending in the presence of bacteria/rate of bending in the absence of bacteria] ×100. Three independent replicates were performed (*n* = 30 number of animals per replicate).

### 4.6. Assay for Ethanol Avoidance Behavior

This assay was performed as described before [26]. Briefly, ethanol avoidance behavior was conducted on an assay plate. The assay plate was prepared by quartering a 9 cm NGM plate into four quadrants: top left (A), top right (B), bottom left (C), and bottom right (D). At the center of the plate, an inner circle with a 0.5 radius was margined. After 6-OHDA exposure and 72 h of HLEA-P3 treatment, 50–100 worms were washed by M9 buffer and dropped on the center of the assay plate. Then, 50 μL ethanol was added into ethanol quadrants (B and C). 50 μL M9 buffer was added into the control quadrants (A and D). The worms were allowed to move for 30 min at 25 °C, and the ethanol avoidance index was calculated using the following formula: Ethanol avoidance index = [(number of worms in control quadrants) − (number of worms in ethanol quadrants)]/total number of worms. Worms that did not move across the inner circle were excluded from the calculation. Three independent replicates were performed (*n* ≥ 50 number of animals per replicate).

### 4.7. Quantitative Analysis of α-Synuclein Aggregation

Aggregation of α-synuclein was assessed using α-synuclein-YFP transgenic worm, NL5901 strain. Synchronized L3 larvae were cultured in OP50-seeded FUdR plates containing 1, 5, 25 and 50 μg/mL of HLEA-P3 and incubated at 20 °C for 72 h. In the untreated control group, worms were fed with OP50 mixed with 1%DMSO (*v*/*v*). After 72 h of treatment, worms were washed by M9 buffer and used for fluorescence imaging. Worms were transferred to a 2% agar slide and anesthetized using 30 mM sodium azide. Then, fluorescence imaging of the whole worms was monitored under a fluorescence microscope (BX53; Olympus Corp., Tokyo, Japan), and the intensity of YFP-tagged human α-synuclein was quantified using Image-J software (National Institute of Health, NIH, Bethesda, MD, USA).

### 4.8. Assay for Thrashing Behavior

Thrashing behavior was monitored using protocol reported in previous study [61]. N2 wild-type and NL5901 PD worms were cultured in OP50-seeded FUdR plates containing 5 and 25 μg/mL of HLEA-P3 until they reached 3 day and 5 day of adulthood. After treatment, adult day 3 and day 5 worms were washed by M9 buffer and transferred to assay plate which is an NGM containing M9 buffer. To avoid overstimulation, worms were settled for 1 min before video recording their body bendings for 30 s. Then, thrashing rate was analyzed using wrMTrck plugin for ImageJ (National Institute of Health, NIH, Bethesda, MD, USA) and represented as body bending per second (bbps).

### 4.9. Quantitative Analysis of Lipid Accumulation

Lipid depositions in NL5901 were performed by mixing *E. coli* OP50 with Nile red, a dye for staining lipid droplets. Firstly, a stock solution of Nile red (0.5 mg of Nile Red in 1 mL of acetone) was prepared and kept at 4 °C. Then, Nile red was diluted in *E. coli* OP50 in a ratio of 1:250. Synchronized NL5901 L3 worms were transferred to OP50/Nile red-seeded FUdR plates containing 5 and 25 μg/mL of HLEA-P3 and incubated for 72 h. OP50/Nile Red mixed with 1%DMSO (*v*/*v*) was used for untreated control. After 72 h of incubation, worms were collected for fluorescence microscopy. Worms were washed three times by M9 buffer and put onto a 2% agar slide. Then, worms were anesthetized by 30 mM sodium azide and enclosed with a coverslip. Fluorescence images were taken using a fluorescence microscope (BX53; Olympus Corp., Tokyo, Japan). The lipid deposition was quantified by measuring the Nile red fluorescence intensity of whole body of worms using Image-J software (National Institute of Health, NIH, Bethesda, MD, USA).

### 4.10. Quantitative Analysis of Intracellular ROS

After exposure to 6-OHDA and treatment with 5 and 25 μg/mL of HLEA-P3 for 72 h, intracellular ROS levels of worms were measured using H_2_DCF-DA probe as reported in the previous study with minor modifications [62]. Briefly, worms were collected and washed three times using M9 buffer. Then, 30 worms were transferred into wells of 96-well black plates containing 50 μL M9 buffer (5 worms per well, 6 wells for each treatment group). A 50 μL of H_2_DCF-DA (final concentration, 25 μM in M9) was added into wells, and the assay plate was incubated at room temperature, in the dark for 1 h. Non-fluorescence H_2_DCF-DA molecules can be converted to fluorescence DCF molecules by intracellular ROS. Therefore, the intracellular ROS was quantified by measuring DCF fluorescence signal using a microplate Fluorescence reader Tecan Spark 10M with excitation at 485 nm and emission at 530 nm. The experiment was performed in three independent trials.

### 4.11. Lifespan Analysis

Briefly, synchronized L3 larvae of 6-OHDA treated worms and NL5901 worms were transferred to OP50-seeded FUDR plates containing 5 and 25 μg/mL of HLEA-P3 and incubated at 20 °C. The number of alive and dead worms was recorded daily. Gentle tapping the plate was applied to determine whether each worm is alive or dead. If the worm moves or shows pharyngeal pumping movement, that worm was counted as alive worm. If the worm lacks movement or pharyngeal pumping, it was assigned as dead worm. Worms with internal organ expulsion or worms that crawled off the plate were recorded as censor worms and excluded from the calculation. Alive, dead and censor worms were counted daily until all worms died. In each treatment group, *n* ≥ 30 per replicate was analyzed. Lifespan assay was performed in three independent replicates. The data were plotted as a survival curve and were analyzed and compared using log-rank (Mantel-Cox) test.

### 4.12. Quantitative RT-PCR

After 72 h of treatment, total RNA of worms was extracted with RNA extraction kit (Qiagen, Hilden, Germany), measured by NanoDrop™ 2000/2000c spectrophotometer (Thermo Scientific, Waltham, MA, USA) and kept at −80 °C. Next, total RNA was converted into complementary DNA (cDNA) using iScript^TM^ Reverse transcriptase Supermix for qRT-PCR (Bio-Rad, Hercules, CA, USA) following the manufacture’s protocol. Then, RT-qPCR was performed using SsoFast™ EvaGreen^®^ Supermix with Low ROX qRT-PCR (Bio-Rad, Hercules, CA, USA) and kept at −20 °C. The qRT-PCR primers specific for *gst-4*, *gst-10*, *gcs-1*, *cat-2*, *bec-1*, *agt-7* and *lgg-1* were selected for the study and shown in Table 2. The qRT-PCR samples were holded at 95 °C for 30 s, followed by 44 cycles of denaturing (95 °C for 5 s) and annealing processes (60 °C for 30 s). After 44 cycles, the samples were then heated up to 95 °C to stop the reaction. EvaGreen fluorescence was detected by Real-time PCR detection system (Bio-Rad, Hercules, CA, USA) and Cq values were obtained. The Cq values of control and treated groups were then calculated via 2^−(ΔΔCq)^ method representing fold change in the expression of each gene. Relative mRNA expression levels were normalized using reference internal control gene, *act-1*. Approximately 800–1000 worms were used for each group. The experiments were performed in triplicates.

### 4.13. Statistical Analysis

All experiments were performed in three independent replicates. The data were statistically analyzed by GraphPad Prism Software (GraphPad Software, Inc., Jolla, CA, USA). Statistically significant differences between treatment groups and untreated groups (1%DMSO) were compared using one-way ANOVA following Tukey–Kramer test for multiple comparisons. *p* value < 0.05 was regarded as statistically significant.

## 5. Conclusions

The present study demonstrated that *H. leucospilota*-derived palmitic acid attenuated loss of DAergic neurons, improved dopamine-dependent behaviors and rescued lifespan in 6-OHDA-induced *C. elegans* PD model. In addition, *H. leucospilota*-derived palmitic acid decreased α-synuclein aggregation, improved motor deficit and prolonged lifespan of worms expressing α-synuclein. Therefore, this study provides evidence that palmitic acid isolated from *H. leucospilota,* has anti-Parkinsonian potential. However, its precise mechanisms and optimal concentration of palmitic acid intake need to be further investigated for development as a nutritional therapy for PD.

## Figures and Tables

**Figure 1 marinedrugs-21-00141-f001:**
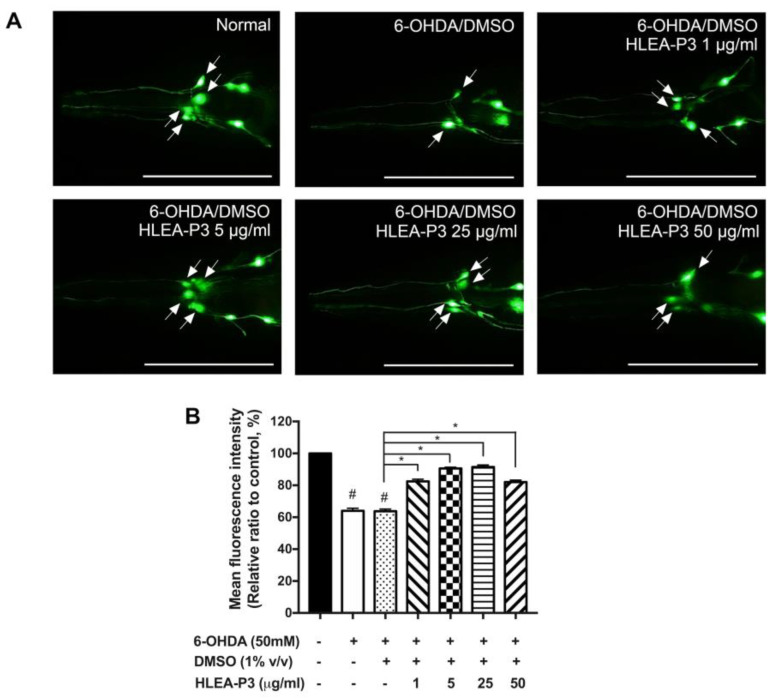
HLEA-P3 attenuated DAergic neurodegeneration induced by 6-OHDA in *C. elegans* model. (**A**) Representative fluorescence images of GFP-tagged CEP neurons (arrows) of normal BY250, 6-OHDA/DMSO-treated BY250 and BY250 exposed to 6-OHDA and treated with HLEA-P3 at doses of 1, 5, 25 and 50 μg/mL. (**B**) Graphical representations for the relative fluorescence intensity of GFP-tagged CEP neurons. The data are presented as a mean ± SEM (*n* = 30, number of animals). The hash (#) indicates a significant difference between normal and 6-OHDA-treated groups (*p* < 0.05). The asterisk (*) indicates significant differences between the untreated group (6-OHDA/DMSO) and HLEA-P3-treated groups at *p* < 0.05. Scale bar is 100 μm.

**Figure 2 marinedrugs-21-00141-f002:**
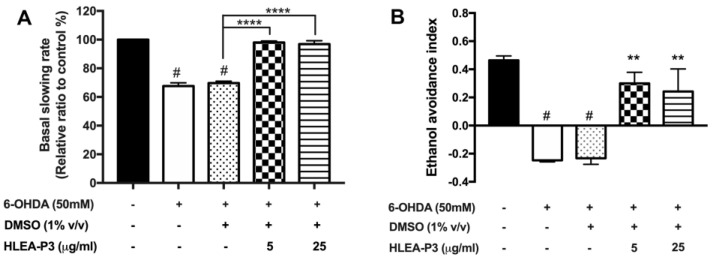
HLEA-P3 recovered dopamine-dependent behaviors in 6-OHDA-treated *C. elegans*. Graphical representations of relative basal slowing rate (**A**) ethanol avoidance index (**B**) of normal, 6-OHDA-induced worms and 6-OHDA-induced worms and those treated with HLEA-P3. The data are presented as mean ± SEM. The hash (#) indicates a significant difference between normal and 6-OHDA-treated groups (*p* < 0.05). The asterisk (*) indicates significant differences between the untreated group (6-OHDA/DMSO) and HLEA-P3-treated groups, ** *p* < 0.01, **** *p* < 0.0001.

**Figure 3 marinedrugs-21-00141-f003:**
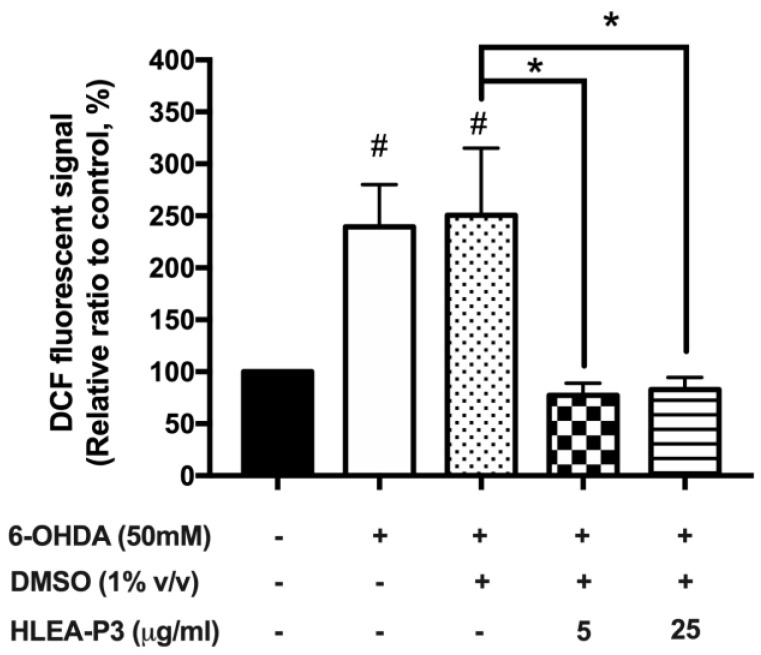
HLEA-P3 reduced oxidative stress in 6-OHDA-treated *C. elegans*. Graphical representations for relative changes in intracellular ROS levels using H_2_DCF-DA. The data are presented as mean ± SEM (three independent replicates, *n* = 30 number of animals per replicate). The hash (#) indicates a significant difference between normal and 6-OHDA-treated groups. The asterisk (*) indicates significant differences between the untreated group (6-OHDA/DMSO) and HLEA-P3-treated groups, * *p* < 0.05.

**Figure 4 marinedrugs-21-00141-f004:**
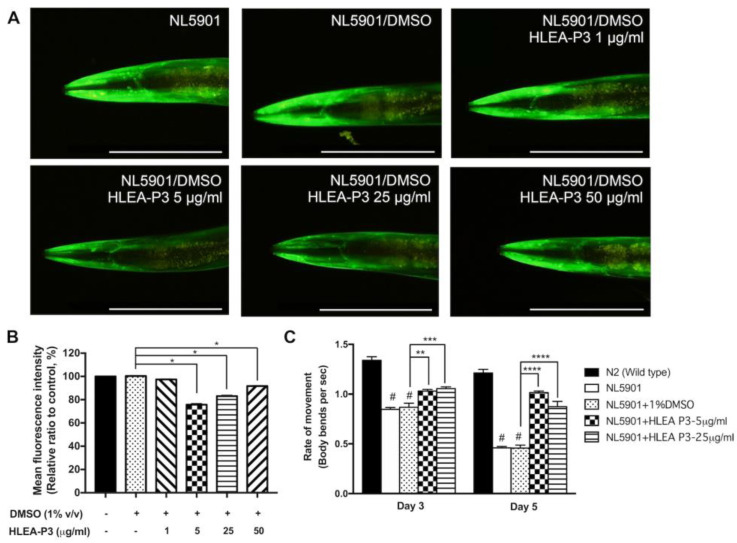
HLEA-P3 reduced α-synuclein aggregation and improved motor deficit in transgenic *C. elegans* NL5901 strain. (**A**) Representative fluorescence image of YFP-tagged α-synuclein expression in body wall muscle of NL5901 strain, NL5901 treated with DMSO, and NL5901 treated with various doses of HLEA-P3. (**B**) Graphical representations for relative fluorescence intensity of YFP expression. The data are presented as mean ± SEM (*n* = 30, number of animals). (**C**) Graphical representations for thrashing rates of wild-type worms, NL5901 and NL5901 treated with HLEA-P3. The data are presented as mean ± SEM (three independent replicates, *n* = 30 number of animals per replicate). The hash (#) indicates a significant difference between wild-type N2 and NL5901 worms (*p* < 0.05). The asterisk (*) indicated significant differences between the untreated group (DMSO) and HLEA-P3-treated group at * *p* < 0.05, ** *p* < 0.01, *** *p* < 0.001, **** *p* < 0.0001.

**Figure 5 marinedrugs-21-00141-f005:**
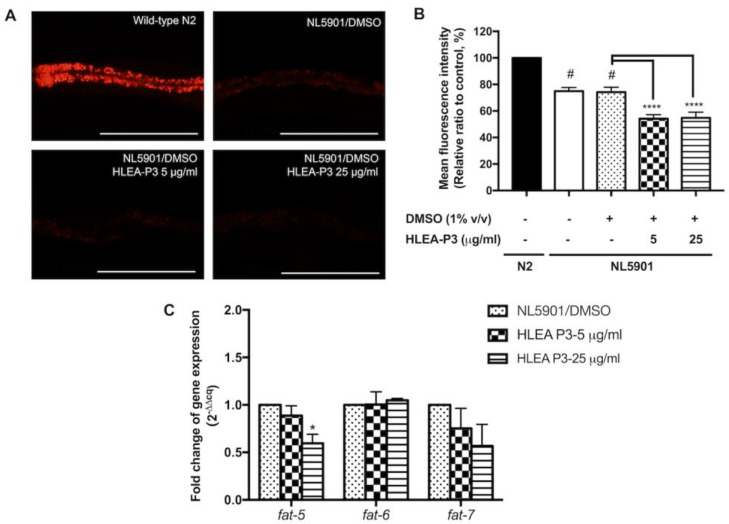
HLEA-P3 modulated lipid deposition in transgenic *C. elegans* NL5901 strain. (**A**) Representative Nile red fluorescence images of wild-type N2, *C. elegans* NL5901 and *C. elegans* NL5901 treated with HLEA-P3 at 5 and 25 μg/mL. (**B**) Graphical representation for Nile red fluorescence signal. The data are presented as mean ± SEM (with three independent replicates, *n* ≥ 30 number of animals per replicate). (**C**) Graphical representation for fold changes of fatty acid desaturase genes: *fat-5*, *fat-6* and *fat-7*. The data are presented as mean ± SEM (with three independent replicates, *n* = 800–1000 number of animals per replicate). The hash (#) indicates a significant difference between wild-type N2 and NL5901 worms (*p* < 0.05). The asterisk (*) indicates significant difference between the untreated group (DMSO) and HLEA-P3-treated groups, * *p* < 0.05, **** *p* < 0.0001. Scale bar is 200 μm.

**Figure 6 marinedrugs-21-00141-f006:**
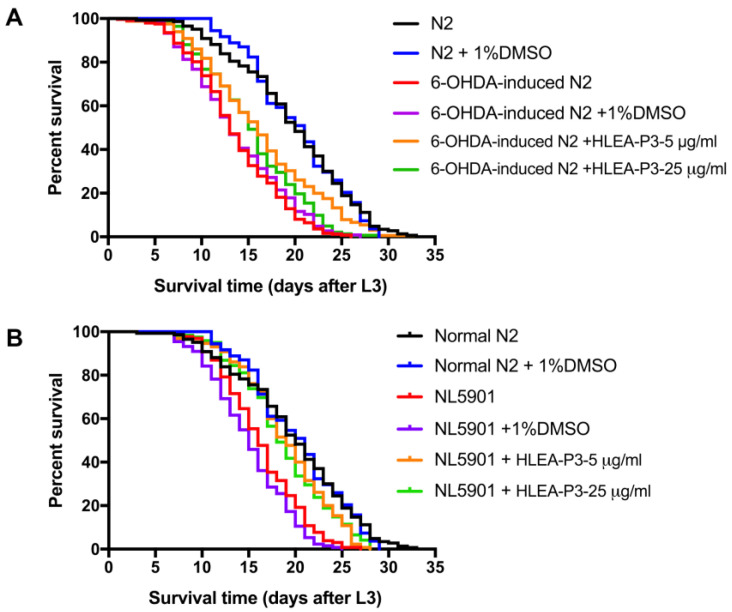
HLEA-P3 improved lifespan of *C. elegans* PD models. (**A**) Kaplan–Meier survival curves of normal worms, 6-OHDA-induced worms and 6-OHDA-induced worms treated with 5 and 25 μg/mL HLEA-P3. (**B**) Kaplan–Meier survival curves of normal worms, NL5901, and NL5901 treated with 5 and 25 μg/mL HLEA-P3.

**Figure 7 marinedrugs-21-00141-f007:**
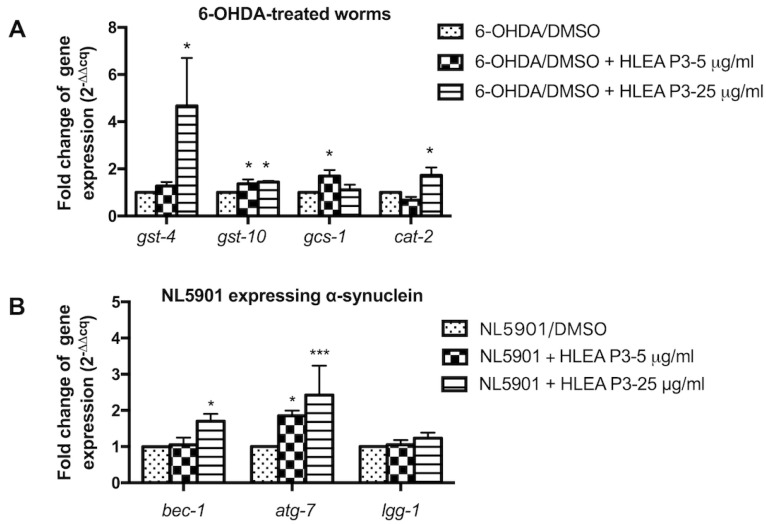
(**A**) Graphical representations of fold changes of mRNA expression levels of antioxidant genes in 6-OHDA-treated worms treated with HLEA-P3. (**B**) Graphical representations of fold changes of mRNA expression levels of autophagic genes in NL5901 worms treated with HLEA-P3. The data are presented as mean ± SEM (three independent replicates, *n* = 800–1000 number of animals per replicate). The asterisk (*) indicates significant differences between the untreated group (DMSO) and HLEA-P3-treated group, * *p* < 0.05, *** *p* < 0.001.

**Table 1 marinedrugs-21-00141-t001:** Mean lifespan, maximum lifespan, number of worms, percentage of increased lifespan and significant *p* values of normal worms (N2) and PD worms and PD worms treated with HLEA-P3.

Treatment	Mean Lifespan ± SD (Days)	Maximum Lifespan (Days)	% Increase Lifespan	Number of Worms	*p* Value (Log-Rank Test)
N2	19.76 ± 0.51	33	-	143	-
N2 + 1% DMSO	20.20 ± 0.49	29	-	108	-
N2 + 50 mM 6-OHDA	13.61± 0.31	26	−31.12	248	####, *p* < 0.0001
N2 + 50 mM 6-OHDA + 1%DMSO	13.75 ± 0.35	27	−30.41	224	####, *p* < 0.0001
N2 + 50 mM 6-OHDA + HLEA-P3 5 μg/mL	16.36 ± 0.49	33	18.98	165	****, *p* < 0.0001
N2 + 50 mM 6-OHDA + HLEA-P3 25 μg/mL	15.36 ± 0.43	29	11.71	142	*, *p* = 0.0105
NL5901	16.28 ± 0.37	27	−17.61	130	####, *p* < 0.0001
NL5901 + 1% DMSO	15.02 ± 0.36	25	−23.99	133	####, *p* < 0.0001
NL5901 + HLEA-P3 5 μg/mL	18.92 ± 0.43	28	25.97	130	****, *p* < 0.0001
NL5901 + HLEA-P3 25 μg/mL	18.71 ± 0.44	28	24.57	122	****, *p* < 0.0001

The hash (#) indicates a significant difference between normal N2 and PD models, ^####^
*p* < 0.0001. The asterisk (*) indicates significant difference between the untreated group (DMSO) and HLEA-P3-treated groups, * *p* < 0.05, ***** p* < 0.0001.

**Table 2 marinedrugs-21-00141-t002:** Primer sequences used for quantitative RT-PCR analysis.

Gene Name	Forward Primer (5′ to 3′)	Reverse Primer (5′ to 3′)
Antioxidative system		
*gst-4*	CCCATTTTACAAGTCGATGG	CTTCCTCTGCAGTTTTTCCA
*gst-10*	GTCTACCACGTTTTGGATGC	ACTTTGTCGGCCTTTCTCTT
*gcs-1*	AATCGATTCCTTTGGAGACC	TGTTTGCCTCGACAATGTT
Dopamine synthesis		
*cat-2*	AAAGCGTGTGAAACGTCAGT	TCTGTCCGACTCCTTTCTCCT
Protein degradation pathway		
*bec-1*	AGATCTCAAAGCTGCGTGTG	AAAAGGCAGAATTCCAGCAGA
*atg-7*	TCTGCAGGATGGATGGTTCG	CTCGGCAAGGTCCATGTGTA
*lgg-1*	AATGGAAACCCAAAGCCCCT	AGGGGAGAAGAGCAACTTCG
Fatty acid desaturation		
*fat-5*	GCCCTCTTCCGTTACTGCTT	CTCCGACTGCCGCAATAGAT
*fat-6*	GCGCTGCTCACTATTTCGGATGG	GTGGGAATGTGTGATGGAAGTTGTG
*fat-7*	CATGGAGGCAAACTCGACCT	GTGGCGTGAAGTGTGAAACA
Housekeeping gene		
*act-1*	ATCGTCACCACCAGCTTTCT	CACACCCGCAAATGAGTGAA

## Data Availability

The data supporting the conclusion in this study are available on request from the corresponding author.

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
