# Peer review of "Anti-Parkinson Effects of Holothuria leucospilota-Derived Palmitic Acid in Caenorhabditis elegans Model of Parkinson’s Disease"

_marinedrugs, 2023, doi:10.3390/md21030141_

Round 1

Reviewer 1 Report

In the subjected manuscript the authors study the anti-Parkinson activity of one of the fractions obtained from the Holothuria leucospilota extract on the Parkinson disease. In this terms the work seems to be a continuation of their previous work concerning another fraction and another active compound. In both papers (submitted and Frontiers in Pharmacology 2022, 13, 1004568) the authors utilize very similar methods what, in general, could be regarded as an advantage showing that they have the necessary know-how to carry such type of investigation. The authors use numerous tests and observe consistent and convincing results. However they do not use any control to show that the observed effects are caused by palmitic acid (as stated). The authors use the term Holothuria leucospilota-derived palmitic acid. What does it exactly mean? Is the Holothuria leucospilota-derived palmitic acid any different from „regular” palmitic acid? The problem with the extracts from natural sources is that they rarely contain 100% pure, single compound. Are the authors sure that their P3 fraction contains only palmitic acid? The NMR spectrum presented in the manuscript suggests so but the authors should provide other analyses to confirm their assignment.

Also the controls with the use of commercial palmitic acid should be included into experiments to draw final conclusions. Otherwise the observed results could be due to the presence of minute amounts of other, undetected compounds.  

Other remarks

- Careful proof-reading of the manuscript is necessary. There are typos in e.g

- the name of the author (Krai meemon);

- (line 136) trteatments (of note – in my opinion singular would be better form here)

- (lines 151, 152) Given that 6-OHDA exerts neurotoxicity by targeting mitochondria and generating oxidative stress [25]. In this study, we investigated the effect…

either the first sentence is not finished or, the most likely, the full stop is not necessary  and the first and second sentence should be joined into one.

-          (line 267) activartion – activation?

- In numerous sentences singular or plural nouns have the wrong verb form (was/ were), e.g:

- (lines 104, 105) The results showed that the GFP-tagged CEP intensity of 6-OHDA-treated and 6-OHDA/DMSO-treated worms were significantly decreased.

- (lines 107,108) Interestingly, the fluorescence intensity of GFP-tagged CEP neurons were significantly increased

- (lines 345, 346) Then, the eluates were examined by the TLC to finally obtain 3 subfractions which then was subjected

- (line 408) Age-synchronized population were

 - (line 527) intracellular ROS levels of worms was measured using..

- (lines 562,563) After 44 cycles, the sample were then heated up to 95 °C to stop the reaction. EvaGreen fluorescence were detected by Real time PCR

- What do authors mean by “The 1H-NMR/13C NMR spectra showed the white powder (lines 287, 288)?

- What is „selective loss of DAergic neurodegeneratio in C. elegans” (line 443)

- The basal slowing rate = 100 – locomotory rate (%), when (where???) the locomotory rate (%) (lines 470, 471)

- After exposure to 6-OHDA and treated (treatment???) with 5 and 25 μg/ml of HLEA-P3 (line 526)

- The description of the NMR spectrum could be improved (lines 291-295)

As shown in Figure 7, Peak A at 0.86 ppm indicated (indicates?) a presence of terminal methyl group (CH3) bouded (???) to the C15. Peak B at 1.23 ppm displayed (??) (corresponds to???) a long chain of methylene protons (CH2) of the C4-C15 atoms. Peak C at 1.61 ppm is related to 2 protons bound to the C3 atom. Peak D at 2.32 ppm is corresponded with (corresponds to the) methylene protons (CH2) of C2.

- The description of the extraction and isolation of the HLEA-P3 fraction seems to be a bit messy.

 - Some information such as the amount of starting material (freeze-dried samples of H. leucospilota) or the reference to the original protocol for the isolation are missing.

- In the sequence word TLC seems to be missing:

In this study, CC was performed using Merck silica gel 60 (finer than 0.063 mm) and Pharmacia Sephadex LH-20. TLC (???) was carried out using Merck precoated silica gel 60 F254 plates

- What do authors mean by: using Sephadex LH-20 condition MeOH 100% (line343, 344)?

Author Response

Dear reviewer,

We would like to thank you very much for comments and suggestions which help us to improve our manuscript. We have edited and revised the manuscript accordingly as highlighted in red in the revised manuscript, and also addressed all of the comments in the file attached.

Sincerely yours,

Krai Meemon

Reviewer 2 Report

In this manuscript, the authors investigated the anti-Parkinson mechanisms of Holothuria leucospilota-derived palmitic acid in neurotoxin-treated and transgenic C. elegans. Overall, this is a well-written manuscript, and much work has been done to generate interesting data. I would recommend the publication of this paper after the following minor revisions.

1.     Line 16: “crude extract from ethyl acetate fraction” – Please recheck the meaning of this description. It seems to say that a crude extract has been prepared from an ethyl acetate fraction (less crude). This seems confusing/does not make sense.

2.     Line 19: “Caenorhabditis elegans (C. elegans)” – By convention, it is unnecessary to introduce “C. elegans”, unlike other uses of abbreviations.

3.     Lines 38 and 47 – The two descriptions of the pathological hallmarks of PD could be combined.

4.     Line 77 (INTRODUCTION): “… HLEA-P3 compound which was later identified as palmitic acid has been found in HLEA fraction” – This statement is confusing. It seems to suggest that the info was previously published. But later in the paper, the authors reported that they identified the compound only in this paper. If it is a result from this paper, I think the authors should consider revising the statement or removing it from the INTRODUCTION.

5.     Line 91 – It would be best to introduce the YFP abbreviation (yellow fluorescent protein) the first time it is mentioned.

6.     RESULTS:

The statistical analysis performed by the authors could be improved by incorporating suitable post-hoc tests following ANOVA, e.g., Tukey HSD, Fisher's Least Significant Difference test, or Duncan's new multiple range test. Then the authors will not be restricted to comparing just the untreated group (6-OHDA/DMSO) and a single HLEA-P3-treated groups each time (see Figures 1B, 2A, 3, 4B, 4C, 5B, 7 and Table 1). They will also be able to simultaneously compare the different levels of HLEA-P3 concentrations applied. Subsequently, this will allow the authors to objectively determine whether there is any dose-dependent change in the parameters assessed.

7.     RESULTS:

Without statistical confirmation, just checking visually, I think dose-dependent changes were mostly not observed in the results presented. For example, in Figure 1, 50 µg/ml of HLEA-P3 produced the lowest fluorescence intensity compared with 1, 5 and 25 µg/ml. Whether there is statistical significance among 1, 5 and 25 µg/ml is also uncertain. In Figure 2, there seems to be no difference in the effects of 5 and 25 µg/ml HLEA-P3. I would recommend that the authors run suitable post-hoc tests to better analyze their data. Then if it is confirmed that dose-dependent changes cannot be found, please discuss a possible explanation for it in DISCUSSION too. Without dose-dependent effects, would it be still convincing to propose HLEA-P3 as a potent anti-PD compound? What future experimental approaches can be applied to resolve this issue?

8.     RESULTS: Some bar charts show SD, whereas others show SEM. Please standardize it. For example, please see Figure 1B vs Figure 2.

9.     The CONCLUSION paragraph may need to be revised. It sounds more like DISCUSSION.

Author Response

(The authors gave the same response as above.)

Reviewer 3 Report

The current manuscript cannot be accepted for publication in its current form. The following issues should be covered

1- The abstract needs careful revision.

Remove lines 16 and 17, the part regarding previous work. More about the experiment design, the tested concentration, and collective sentences about the impact of the study should be added.

In keywords. Dopaminergic neuron; should not be italicized. Add sea cucumber to the keywords. 

Palmitic acid is a well-known compound for a long time ago. Move the data for the compound characterization to the supplementary materials. Also, add all spectral data, NMR and HRMS. 

A list of abbreviations should be included.

Improve the quality of the figures since some of them are unreadable.

English editing is needed, there are many typing and grammatical mistakes. Remove the pronoun we and rephrase the sentences.

The conclusion, add future perspective and recommendation.

A diagram illustrating the possible mechanism of action of palmitic acid should be included in the conclusion.

Author Response

(The authors gave the same response as above.)

Round 2

Reviewer 1 Report

After the revision the manuscript is improved, however I still raise a question of the appropriate controls. I agree with the authors that they used proper negative control. I also acknowledge the information that the identity of the "compound 3" from the HLEA extract was revealed at later stages of the research. But, in my opinion the results should be repeated with the use of pure palmitic acid. Otherwise the observed effect might be exerted by other compounds present in the fraction (the synergetic effect?). On the mass spectrum provided in the Supplementary data other compounds are present. Why the authors do not discuss this results?

Author Response

Dear reviewer,

We thank you very much for your further comments and concern, which help us to improve our manuscript. Please find our responses to your comments and suggestion in the file attached.

Best regards,

Krai Meemon

Reviewer 3 Report

No comment

Author Response

Thank you the reviewer for the positive outcome of our revised manuscript.